# Viral Membrane Fusion Proteins and RNA Sorting Mechanisms for the Molecular Delivery by Exosomes

**DOI:** 10.3390/cells10113043

**Published:** 2021-11-05

**Authors:** Ilya Zubarev, Dmitry Vladimirtsev, Maria Vorontsova, Igor Blatov, Konstantin Shevchenko, Svetlana Zvereva, Evgenii A. Lunev, Evgeny Faizuloev, Nikolay Barlev

**Affiliations:** 1Faculty of Medicine, Lomonosov Moscow State University, 119991 Moscow, Russia; maria.v.vorontsova@mail.ru; 2Laboratory of Intracellular Signaling, Moscow Institute of Physics and Technology, 141701 Moscow, Russia; Vladimirtsev.da@phystech.edu (D.V.); Igblatov@gmail.com (I.B.); sv.zvereva.2014@gmail.com (S.Z.); 3The National Medical Research Center for Endocrinology, 117292 Moscow, Russia; 4Institute of Cytology RAS, 194064 St. Petersburg, Russia; konst.shevchenko@gmail.com; 5Chumakov Federal Scientific Center for Research and Development of Immune-and-Biological Products of Russian Academy of Sciences, 108819 Moscow, Russia; 6Institute of Biomedical Chemistry, 119121 Moscow, Russia; 7Institute of Gene Biology, Russian Academy of Sciences, 119334, Moscow, Russia; lunev357@gmail.com; 8I. Mechnikov Research Institute of Vaccines and Sera, 105064 Moscow, Russia; faizuloev@yahoo.com

**Keywords:** fusion protein, extracellular vesicles, target delivery, RNA sorting

## Abstract

The advancement of precision medicine critically depends on the robustness and specificity of the carriers used for the targeted delivery of effector molecules in the human body. Numerous nanocarriers have been explored in vivo, to ensure the precise delivery of molecular cargos via tissue-specific targeting, including the endocrine part of the pancreas, thyroid, and adrenal glands. However, even after reaching the target organ, the cargo-carrying vehicle needs to enter the cell and then escape lysosomal destruction. Most artificial nanocarriers suffer from intrinsic limitations that prevent them from completing the specific delivery of the cargo. In this respect, extracellular vesicles (EVs) seem to be the natural tool for payload delivery due to their versatility and low toxicity. However, EV-mediated delivery is not selective and is usually short-ranged. By inserting the viral membrane fusion proteins into exosomes, it is possible to increase the efficiency of membrane recognition and also ease the process of membrane fusion. This review describes the molecular details of the viral-assisted interaction between the target cell and EVs. We also discuss the question of the usability of viral fusion proteins in developing extracellular vesicle-based nanocarriers with a higher efficacy of payload delivery. Finally, this review specifically highlights the role of Gag and RNA binding proteins in RNA sorting into EVs.

## 1. Introduction

Recent developments in precision medicine are largely due to the successful implementation of specific delivery of biologically active cargos to the target tissues. This should significantly decrease the possible side effects of therapeutic agents, while maintaining an appropriate efficacy. The successful implementation of this approach into clinical use would be beneficial for the treatment of a broad spectrum of various diseases, ranging from cancer, to bone fractures, to monogenic diseases. Targeted delivery may be implemented by using various carriers such as viral vectors, liposomes, nanoparticles, extracellular vesicles (EVs) and other artificial vectors. Natural carriers such as viruses or EVs provide an effective alternative to artificial nano-carriers, as they have innate tropism to particular tissues. The best example of such carriers is recombinant non-replicating viral vectors. They have been extensively studied as promising tools for transgene delivery. Depending on the serotype, they possess a natural tropism to different organs and tissues [1]. Particular strains of adeno-associated viruses (AAVs) were used in clinics for tissue-specific delivery to the intestines, pancreas, salivary glands, and maxillary sinus [2]. However, the capsid proteins may trigger intracellular and extracellular antiviral responses [3]. To this end, their immunogenicity may be reduced by using the methods of genetic engineering and synthetic biology. These achievements open up new possibilities for in vivo and in vitro delivery systems [4,5,6].

Notwithstanding the recent advances in the development of new biomaterials and nanofabrication, there are still many problems to be solved before the next generation of “smart” nanocarriers may be applied in everyday clinical practice. Their surface should be modified appropriately by various targeting molecules that would confer the affinity of nano-vehicles for specific tissues. One of the major obstacles is that higher vertebrates have various barriers that prevent the invasion of foreign materials, including nanosized delivery systems. Immune cells efficiently recognize and phagocyte the foreign biomaterial. At the same time, blood enzymes alter the structure and biological properties of these exogenous particles [7]. Although the small size of nanoparticle vectors facilitates their permeability, it also leads to delayed toxic effects [8,9]. Besides, they have poor site-specific accumulation, and are incapable of effective penetration through the tumor microenvironment [10]. Even when delivered to the target organ, the payload will not necessarily be able to efficiently enter the cells.

Many studies in this field have been focused on the development of lipid-based systems such as liposomes and lipid nanoparticles. One of the main problems with such systems is their failure to naturally fuse with cell membranes. Usually, they enter a cell via an endocytosis-based pathway. Natural carriers such as exosomes, on the contrary, have a complex surface composition with a specific set of membrane proteins that assist efficient targeting and entrance into the cells. When enriched with small non-coding RNAs (including miRNAs), mRNAs, fragments of DNA, or proteins [11,12,13], exosomes can be employed for targeted gene repression and cell reprogramming [14]. It has been shown that exosomes can be used as therapeutic delivery systems to treat various pathologies [15]. However, exosomes themselves do not actively promote membrane fusion, which is an important step for the efficient delivery of the drug payload into the target cell [16].

The most common way for nanoparticle vectors to enter the cell is via the endosomal pathway. In this case, loaded therapeutic molecules interact with the endosomal, and then the lysosomal, environment. The latter is characterized by high acidity and high protease activity, which may alter the structure or properties of the delivered therapeutic molecules [17]. The endosomal pathway can be bypassed via fusion with the target cell; however, this process requires the breaking of the energy barrier that prevents the fusion of two lipid bilayers of interacting biological systems [18,19]. Even close and stable contacts between membranes under physiological conditions usually do not lead to membrane fusion. Many attempts have been made to increase the specificity of the interaction between liposomes or lipid nanoparticles with the target cells by attaching oligopeptides, antibodies or receptors. However, the efficacy of these strategies has been invariably low. The presence of a temporary contact between the two lipid bilayers does not ensure their effective fusion [20]. Altering the lipid composition of membranes or modifications thereof with different energy-decreasing molecules enhances their nonspecific fusion with other membranes [21,22]. Thus, it does not come as a surprise that liposome-based strategies have very limited efficacy for systemic delivery.

Cell to cell communication is carried out through direct interaction, secretion of various soluble factors or different cell vesicles that can have an impact on other cells. Such vesicles regulate fundamental biological properties in multiple ways, merging their membrane contents into the recipient cell plasma membrane and delivering effectors including transcription factors, oncogenes, small and large non-coding regulatory RNAs, mRNAs and infectious particles into recipient cells. Among the various articles considering exosomes as delivery vehicles, the emphasis is usually on the content of vesicles and their distribution throughout the body, but not on the mechanism of their entry into cells [23,24,25]. As with liposomes, exosomes need to bypass the endosomal pathway and avoid degradation in lysosomes, which can be achieved through membrane fusion. Here, we focus on the mechanism of the fusion between the cell and drug-loaded EVs and exosomes. This information may help to design novel approaches for the specific and efficient delivery of the molecular load, in particular miRNA, to manipulate the gene expression in the target cells.

## 2. The Process of Membrane Fusion: Breaking the Energy Barrier

The lipid bilayer of liposomes, lipid nanoparticles, and biological membranes consists of charged polar head surrounded by a hydration layer, and neutral tails oriented towards each other. Regardless of the complexity of the system, the fusion of two membranes is accompanied by the interaction of various surface forces: hydration repulsion, hydrophobic attraction, and the Van der Waals force [26]. When put in proximity, the two lipid bilayers are weakly attracted by a weak Van der Waals force and are strongly repulsed due to the interaction of phospholipid polar groups and the hydrated shells. The fusion is the result of the hydrophobic attraction of the internal hydrophobic groups, which are surrounded by an aqueous environment. When cells form intercellular contacts, the distance between them shrinks until a thin layer of water molecules remains, and membrane fusion does not occur [19]. Each water molecule can form up to four bonds with the surrounding water molecules. Since the water molecules have an affinity for hydrophilic groups of phospholipids, they tend to form an ordered layer of charged molecules on the membrane surface. As a result, two hydrated bilayers of contacting membranes simultaneously experience a strong repulsion and a weak attraction to each other. The “hydration force” increases sharply when the distance between the surfaces of two bilayers becomes less than 20 Å, and the membranes get attracted to each other [27].

The energy barrier to the formation of the water-free lipid conformation is directly proportional to the intermembrane distance. Shortening the distance between the membranes leads to an attenuation of the repulsive forces between them. Membranes must stay in close juxtaposition and become destabilized to allow the transition from the bilayer to the non-bilayer conformation. During the membrane interaction, the lipid bilayer undergoes perturbation and destabilization, which causes high local curvature and membrane fusion [18]. When the membranes are close to each other in the contact area with an infinitely small radius, it leads to the dehydration of their surfaces and the formation of a fusion pore [28]. Next, a water bridge is formed and the internal contents of the two structures fuse together [29]. The process of membrane fusion itself is composed of two stages: hemifusion and pore formation. The hemifusion stage includes the merging of the outer leaflets of the opposing bilayer, whereas inner leaflets do not merge. During pore formation, both outer and inner leaflets fuse, forming a connection between the two integrating compartments [30].

From an evolutionary point of view, membrane fusion has been developed as a well-controlled process, while random fusions could lead to serious problems for the multicellular organism [19,31,32]. Specifically, an uncontrolled fusion may lead to the formation of syncytial organisms, for which multicellularity and cell diversity is impossible. In biological systems, the fusion of membranes is implemented by conformational changes of specific fusion proteins [33]. Enveloped viruses presumably were the first living entities that were able to overcome the high-energy barrier to fuse with biological membranes using special viral fusion proteins (spikes) [34] (Figure 1). The spike typically consists of an outer subunit that binds to the target molecule on the surface of the host cell and a transmembrane subunit which mediates the membrane fusion. The activity of fusion proteins is strongly regulated during the course of infection. For instance, they are inactivated during the biogenesis and transport of the viral particles. The insertion of the fusion protein subunit into the outer leaflet of the target membrane may be exerted in a monomeric or trimeric conformation [35]. The trimerization is mediated by the membrane interaction [36]. Albeit differing from each other in the prefusion form, all post-fusion 3D structures of viral fusion proteins show a trimeric hairpin conformation [33].

The energy barrier for hemifusion is estimated to be ≈50 kcal/mol. Therefore, membranes are unable to fuse under normal conditions [37,38]. In biological systems, this process is accelerated by viral fusion proteins (Harrison 2015). For example, an individual hemagglutinin (HA) fusion peptide has the free binding energy for connecting a lipid bilayer of about 8 kcal/mol. Thus, the pull force of three fusion peptides in the trimeric state is estimated as 24 kcal/mol. Therefore, a triple trimeric HA should generate enough force to overcome the kinetic barrier of hemifusion (50–70 kcal/mol) [39]. Notably, for the West Nile virus and Kunjin virus, the critical number of trimers to overcome the barrier was reported as two [40]. Despite the fact that the common fusion rate depends on the threshold number of adjacent trimers, a correlation between fusion peptide insertion depth and fusion effectiveness has also been demonstrated. A deeper protrusion of the fusion peptide into the target cell membrane results in a more effective fusion [41]. At the intermediate insertion depth of fusion peptides, a periodic opening and closing of a pore in the target membrane is possible. It is the secondary structure of the protein rather than the primary amino acid sequence that determines the depth of its insertion [42]. The trimerization of fusion proteins determines the effective surface abundance of activated monomers in the contact zone between membranes and is the rate-limiting stage of the fusion processes.

The onset of the fusion process initiates conformational changes in the virus fusion protein. A trimeric intermediate brings together the target cell and viral membranes and then folds back to a hairpin conformation. The trimer penetrates the target membrane via a hydrophobic loop domain, which leads to the deformation of the two membranes [43] and to the formation of a lipidic stalk or hemifusion intermediate [44]. The latter is driven by the fold-back process through the energy of protein refolding [45]. In this state, the mixing of the outer membrane leaflets (a stage of hemifusion) is initiated, followed by the opening of a small fusion pore. This leads to a complete compartment fusion [33]. Thus, entry into the cell can be provided by fusion proteins, whereas the correct recognition can be achieved by specific receptors on the surface of target cells.

Several mechanisms for the activation of viral fusion proteins have been described. They include receptor binding (retroviruses, lyssaviruses), receptor binding to a separate attachment protein (paramyxoviruses), the binding of a receptor with a coreceptor (HIV), low pH (alphaviruses, flaviviruses, and influenza virus), and receptor binding under low pH (avian sarcoma/leukosis virus). All these signals trigger the thermodynamically favored process of the refolding of the fusion protein into a stable final post-fusion form [33]. In the first case, the binding of the viral fusion protein to the target cell triggers endocytosis, allowing viruses to enter the endosome. Under a low pH, fusion proteins undergo conformational changes that lead to the fusion with endosomes from the inside [46]. Other activation mechanisms of the fusion proteins likely involve co-receptor(s) located on the surface of the target cells [18]. For example, conformational changes in the HIV-1 gp120 protein triggered by binding to the CD4 receptor and co-receptor (e.g., CCR5 or CXCR4) leads to the refolding of the gp41 protein, and subsequently, to membrane fusion [47].

## 3. Eukaryotic Orthologues of Viral Fusion Proteins

Importantly, similar to the viral fusion proteins described above, eukaryotic cells have their own special proteins that are able to merge membranes. Several genes that are involved in the process of membrane fusion during embryo differentiation and development were originally acquired from viruses. Endogenous retroviral genes are involved in syncytium formation and cell fusion. Inside the syncytium, all cells are connected through channels that have high permeability for macromolecules, allowing them to spread between neighboring cells. In terminal differentiation, the lens fibers are deprived of organelles and form syncytia, while adjacent immature fibers ensure the diffusion of nutrients [48]. During the embryogenesis or regeneration of striated muscle tissue, the precursors of these cells fuse with each other to form a muscle fiber. Muscle formation is assisted by Duf and Sns proteins that exhibit the membrane attachment and fusion activities [49]. The role of fusion proteins in the origin of the sexual reproductive process has also been proposed. Structural and phylogenetic analysis of HAP2 sperm–egg fusion proteins revealed that they are homologous to the fusion proteins of the Zika and Dengue viruses [50]. HAP2 proteins and the class II viral fusogens have similar structural properties as they insert into the host cell membrane and destabilize it at the late stages of gamete fusion [51]. Summing up these facts, it is tempting to speculate that the horizontal transfer of viral fusion genes has contributed to the development of sexual reproduction [52], thereby allowing the formation of complex multicellular organisms [53]. ERV genes exert many functions in the development, including gene activation in a zygote after fertilization [54], promotion of placenta development, protection of the host from infection, and regulation of genome plasticity [55]. The syncytin-2 protein (HERV-FRD) is an immunosuppressant, whose immunosuppressive domain helps the fetus to escape the mother’s immune system [56]. Syncytin-1 (HERV-W) participates in the formation of placental trophoblast, which is in turn is involved in the fusion of cancer cells and human osteoclasts [57]. The viral fusion proteins are deeply integrated into many morphogenetic processes important for the normal body functioning.

## 4. Exploitation of the Host Mechanisms: EVs for Infection

The common way to transmit viruses among the host’s cells is to assemble particles intracellularly and then release them into the extracellular environment. However, during the infection, virus-infected cells can fuse and form syncytia. Furthermore, viruses can exploit existing intercellular communication pathways mediated by EVs to infect new populations of cells [58]. They can also be transported via EVs [59] including exosomes-small (30–100 nm) vesicles that are released by cells into the extracellular space [60].

Exosomes, being part of the cell–cell cargo delivery system, potentially co-evolved together with viruses [61,62]. This fact explains many specific virus–exosome interactions and the presence of mechanisms to involvement in the membrane traffic of eukaryotic cells [63]. Some non-enveloped viral particles are surrounded by cell membranes and form exosome-like vesicles that protect them from the antibody-mediated immune response (see Table 1). Different viral components of EVs may participate in the infection by assisting the targeting, internalization and replication of viruses in the target cells. The B-cell tropism of EBV gp350 in exosomes is thought to be mediated by interacting with the B lineage marker CD21 and has been applied in clinical settings [64]. Neurotropic Zika [65] and rabies viruses [66] stimulate the production of exosomes containing viral RNA and glycoproteins in neurons for further transmission. The tick-borne flaviviruses employ exosomes for the viral RNA and proteins transmission from arthropods to humans [67]. At the same time, viruses incorporate cellular proteins and nucleic acids into the EVs to escape immune response [68]. EVs could transmit viral receptors CXCR4 [69] and CCR5 [70] to healthy cells that were devoid of these receptors, making them susceptible to the infection.

Viruses exploit the Endosomal Sorting Complex Required for Transport (ESCRT) pathway to intercept cell membrane traffic. Viral structural proteins, such as retroviral Gag proteins, arenaviral Z proteins, and filoviral VP40 proteins, act as ESCRT adaptors. They interact with ESCRT components via specific sequence motifs similar to cell proteins [91]. Several specific motifs (P(T/S)AP, PPxY, ΘPxV, YP(x)nL) of the viral structural proteins mediate the recruitment of the ESCRT machinery to the particle assembly sites [92,93,94,95]. In particular, many structural proteins contain the YPXL assembly domains that bind directly to the central V-domain of ALIX, the early acting ESCRT factor [95]. Viruses that contain such YPXL domains include retroviruses, arenaviruses, flaviviruses, hepadnaviruses, herpesviruses, paramyxoviruses, and tombusviruses [95]. Nef is a scaffold protein of HIV-1 which is associated with lipid-raft microdomains during the assembly of retroviral particles [96]. To illustrate the ability of Nef to interact with sorting proteins or RNA and get loaded into exosomes, a non-functional mutant of the HIV-I Nef protein (Nefmut) was used [97]. The Nefmut–GFP fusion protein was successfully loaded into exosomes [98]. Fusing Nefmut to several viral proteins, including Ebola virus VP24, VP40, and NP, influenza virus NP and others, resulted in the expression of stable fusion proteins and their efficient loading into exosomes [99]. Thus, due to specific structural motifs, viral proteins stimulate the formation of EVs to move between cells.

Viral attachment and subsequent cell entry are mediated by a number of membrane molecules located on the surface of the host cell [100]. Several classes of such molecules serve as binding factors or entry receptors that are recognized by fusion proteins of mammalian enveloped viruses (see Table 2). Therefore, it is tempting to speculate that choosing the right viral protein from the broad variety of viral proteins that recognize specific cellular receptors may improve the delivery efficiency of specific transgenes.

Viral pseudogenes are involved in the building of syncytia in muscles, osteoclasts, the lens, placenta, embryonic ovaries, testes, sexual reproduction. However, there is only a limited number of organs with syncytia, which restricts the use of human fusion proteins for the vesicle-mediated targeted delivery. Therefore, the idea of targeting EVs and exosomes with synthetic viral fusion proteins seems to be worthy of experimental testing. If this approach proves efficient, it would help to expand the application and efficacy of these systems by allowing them to escape degradation in the course of the endosomal pathway [157].

Altogether, numerous variants of viral capsid proteins provide a new platform for using viral tropism as a tool for selective tissue-specific recognition and merging with the target cells. As a new approach in cancer immunotherapy, it was proposed to use tumor xenogenization (a process of addition of the pathogenic antigen that increases the visibility of cancer cells for the host immune system) through fusogenic exosomes with special fusogenic viral antigens (e.g., VSV-G). Ideally, this approach would enhance the recognition and uptake by dendritic cells, which would then recruit T-lymphocytes and ultimately result in the suppression of tumor growth [158,159]. The same VSV-G protein was applied as part of virus-mimetic fusogenic exosomes for the insertion of integral proteins of interest into the target membrane [160,161]. The presence of not only VSV-G but also other viral envelope proteins on the EV’s membrane stimulates specific attachment and fusion with target cells [162,163]. For example, the Lamp2b protein fused with the neuron-specific rabies viral glycoprotein (RVG) peptide can efficiently target neurons [164]. EVs with the receptor-binding domain (RBD) of the viral spike protein and siRNAs against SARS-CoV-2 can specifically recognize the ACE2 receptor on the surface of target cells. These EVs can be effectively used in vivo for attachment, fusion and cellular entry during the delivery of potential antiviral agents which act as a potential therapeutic agent against SARS-CoV-2 infection [165]. Human fusion proteins are also involved in the intercellular communication through fusogenic exosomes, for example, in intercellular communication in the placenta [166] and muscles [167].

Another way of increasing the specificity of exosome targeting can be achieved via the insertion of specific integrins into membranes of exosomes. Integrins are heterodimeric cell membrane proteins consisting of α and β subunits. The major biological role of integrins is to mediate cell–cell contacts at specific focal adhesion points. Importantly, integrins transmit transmembrane signals bi-directionally thereby modulating the intracellular events in response to specific ligands. During the formation of exosomes, integrins are captured on their surface. Consistent with their functions, exosomal integrins have been shown to be capable of guiding exosomes to specific tissues. In the pioneering work, Lyden’s group have shown that exosomes derived from breast and pancreatic cancer cells ectopically expressing integrin α6β4 were preferentially targeted to the lung tissue since the latter expresses laminin, the ligand for the α6β4 integrin. On the contrary, those exosomes that expressed integrin αVβ5, which specifically recognizes fibronectin, were preferentially distributed to the liver cells, which are known to abundantly express this ECM protein [168]. Thus, one would predict that an artificial embedding of specific integrins into the membrane of EVs should increase the tropism of exosomes to specific tissues.

## 5. Exploitation of Viral Sorting Mechanisms for RNA Loading into EVs

Exosomes, especially the ones derived from cancer tissues, are deemed to affect the physiology of target cells [169]. It is known that the majority of vesicular RNAs are non-coding RNAs (ncRNAs) including tRNAs, rRNAs, microRNAs, and small nuclear RNAs [170]. ncRNAs act as important mediators of the intracellular communication through the regulation of energy metabolism and cell-to-cell signaling [171]. Therefore, ncRNA-loaded EVs may be of interest from the therapeutic point of view.

Loading of RNA into EVs can occur in several ways: randomly, due to the abundance of certain RNA species in the cytosol or selectively, or by RNA-binding proteins (RBPs) that recognize specific motifs/secondary structures in the RNAs to be sorted. These RBPs recognize specific (GAGAG [172] and GGAG [173]) sequence motifs in miRNA or UTR in mRNA as EV packaging signals. The group of RBPs responsible for sorting RNAs into EVs is heterogeneous and includes many proteins, e.g., AGO2/Argonaute [174], ALIX70 [175], annexin A2 [176], major vault protein (MVP) [177], the human antigen R (HuR) [178], heterogeneous nuclear ribonucleoproteins [173], FMR1 [179], and hnRNPU [180] to name just a few [181].

Viruses possess a unique system of specific recognition and package only the viral RNA into a viral particle despite the fact that there are thousands of mRNAs in the animal cell [182]. This specificity is ensured by the recognition of a special 3′-UTR or 5′-UTR RNA packing signal by Gag proteins [183]. The gag protein binds to the cis-acting RNA element or psi packaging sequence (known as Ψ) in the 5′ untranslated region (UTR) through the NC domain [184,185]. The packaging signal (the psi region) folds into four stem-loop structures (SL1, SL2, SL3, SL4) [186,187]. A minimal 159-nt RNA sequence that includes SL1–SL3 can dimerize and is competent to bind gag NC and is sufficient to induce packaging of an any RNA sequence with the psi signal [188]. Mutagenesis studies suggested that it is the structure of the Ψ hairpin, rather than its sequence is critical for genome dimerization and packaging [189]. The packaging signal consists not merely of the linear RNA sequence of ψ, but also includes the 3-dimensional structure formed upon dimerization. The presence of two packing signals on one mRNA molecule induces packing in the form of monomers [190].

The psi signal of RNA can be recognized by special viral proteins, in particular the gag protein. There are four key domains found in all Gag proteins (listed N- to C-terminus): the matrix (MA) domain, which is primarily associated with membrane-binding capability; the capsid (CA) domain, which mediates numerous Gag–Gag interactions; the nucleocapsid (NC) domain, which is involved in specifically packaging the genome RNA (gRNA); and p6, which contributes to the release of the assembled particle from the host cell by interacting with the cellular endosomal sorting complexes required for transport (ESCRT) machinery [191]. The NC domain specifically interacts with the G-rich and G/U-rich sequence motifs of gRNA through two evolutionarily conserved Cys-His boxes with Zn2+ ions, allowing for high-affinity NC-gRNA interactions [192,193].

Recombinant Gag proteins can assemble into particles in vitro, but this assembly requires the addition of nucleic acids; nearly any single-stranded nucleic acid longer than ~20–30 nucleotides can support assembly under these conditions [194]. Using gRNA of different lengths (3 kb, 8 kb, or 17 kb), it was shown that binding to nucleic acid primes Gag assembly, indicating that viral genome packaging is not regulated by the RNA mass [195]. RNAs can serve as a scaffold for Gag proteins to assemble whereas NC binds to RNA, and this interaction leads to the multimerization of the Gag polyproteins [196]. RNA binding proteins anchor to the plasma membrane, oligomerize and induce the release of vesicles by budding from the plasma membrane [197].

There are ~100 retroviral Gag-like genes in the human genome that can self-assemble into RNA-carrying capsids [198]. Notably, its Gag domain could bind to the UTR of some RNA and drive the extraction of EVs or viral-like particles. It has been suggested that synthetic RNA sorting can be programmed by inserting appropriate retroviral elements. Some retroviral-like Gag proteins (e.g., Arc) can bind RNA and produce EVs by multimerizing into virus-like capsids similar to the viral RNA packaging [199]. Notably, the Peg10 (paternally expressed gene 10) gene, originally of viral origin, plays an important role in the placental development. Nevertheless, Peg10 also maintains the ability to bind and package its own mRNA. This finding allowed to create an artificial packaging system where heterologous RNAs fused with Peg10-3′UTR were efficiently packed by Peg10 [200,201]. Furthermore, using the viral RNA packaging system, it became possible to specifically incorporate sgRNA with the CRISPR-Cas9 nuclease into EVs for genome editing [202]. Thus, an in-depth understanding of the viral genome packaging mechanisms should aid in developing an effective platform for RNA delivery and expand the repertoire of gene therapy applications.

## 6. Conclusions and Perspectives

An understanding of the mechanisms of membrane fusion and RNA sorting promises new approaches to control the loading of EVs with specific RNAs. More specifically, the recognition of the RNA packaging signal by the RNA-binding domain of synthetic viral-like proteins may assist the loading of mRNA and miRNA of interest into the EVs (Figure 2). To this end, it is possible to create a cell line with constitutive co-expression of the viral fusion protein and the RNA binding protein to assist the production of targeted EVs. Such EVs should be biosafe as they will not contain the whole viral genome and will only be weakly immunogenic. However, for therapeutic purposes, a high yield of extracellular vesicle production must be obtained. The production of large quantities of pure extracellular vesicle requires a large amount of primary cell material. Increasing the biogenesis of EVs using special viral proteins in the producer cells is one of the viable approaches to increase the quantities of RNA-containing EVs [160].

Given the fact that the HIV genome (9.8 kb) can be packed within the volume of the exosome size, one should potentially be able to pack 500 copies of miRNAs into a single exosome [203]. Furthermore, enveloped coronaviruses have virion sizes similar to those of EVs and contain single-stranded RNA up to 30 kb in size. This is the largest known genome size of an RNA virus and can serve as a guideline for the maximum RNA capacity of these EVs.

We hypothesize that the expanding knowledge of different mechanisms that viruses utilize to interact with host cells, together with the ability to manipulate and engineer hybrid EVs, could be the basis for the development of future therapies. In fact, effective targeting delivery systems have already been developed by viruses in the course of evolution, which can specifically navigate through the human body. Therefore, the success of future studies relies on the ability to repurpose the already available delivery systems for the benefit of human health.

## Figures and Tables

**Figure 1 cells-10-03043-f001:**
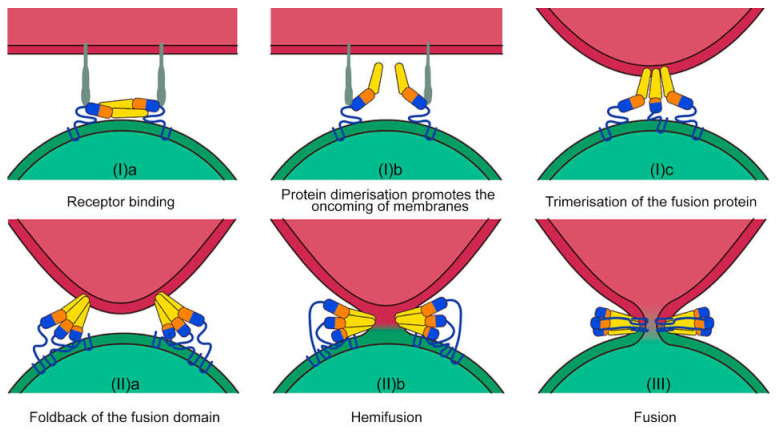
Molecular mechanisms of membrane fusion. Fusion proteins (shown in multicolor monomers) in viral particles (shown in green) recognize the target molecule (shown in grey) on the cell surface (shown in red). A change in the conformation of fusion proteins is required for pore formation. This process includes several stages: membrane recognition (**Ia**), docking (**Ib**) and trimerization (**Ic**), membrane approaching (**IIa**), deformation, destabilization (**IIb**), and fusion pore formation (**III**) and its growth. The latter leads to the merging of the two compartments.

**Figure 2 cells-10-03043-f002:**
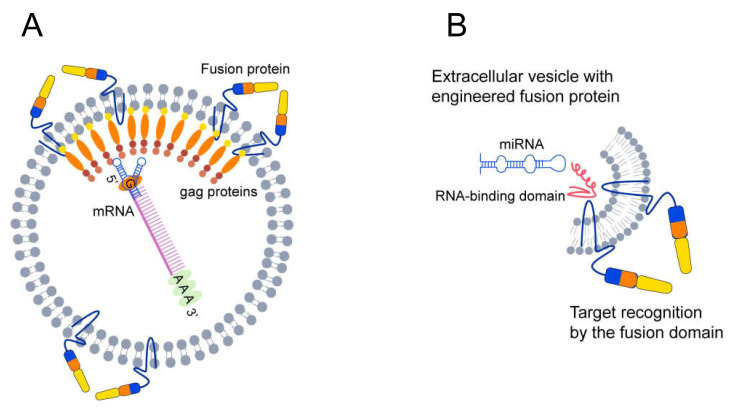
EVs functionalized with fusion proteins. (**A**) Design of a system for sorting mRNA into EVs using viral gag protein and the packing signal. The gag protein selectively binds the packaging signal and saturates the vesicles with mRNA. (**B**) The EV vector may be functionalized with an engineered fusion protein (shown in multicolor monomers) that contains an RNA binding domain (shown in red). The intravesicular domain of fusion proteins consists of an RNA-binding domain (RBD) that can bind specific miRNAs with the packaging signal. The RNA packaging signal is recognized by the RBD of fusion proteins and may stimulate the upload of miRNA into EVs. Fusion proteins may interact with an extravesicular recognition domain and form the structure required for merging with the target membrane.

**Table 1 cells-10-03043-t001:** Viruses that use exosomes for cell-to-cell delivery.

Viruses	Exosome Content	References
Non-enveloped virus (Exosomes-like vesicles):		
Hepatitis A, B, C	viral particles, viral RNA, proteins	[71,72,73,74]
Poliovirus (PV)	virions, viral RNA and replication proteins	[75]
**Enveloped virus:**		
Human immunodeficiency virus (HIV)	virus and viral constituents (such as viral microRNA (miRNA), viral proteins Gag and Nef	[76,77,78,79,80,81,82,83]
dengue virus (DENV)	complete RNA genome and proteins of DENV	[84,85]
Ebola virus (EBOV)	proteins (VP40, GP, NP) and RNA	[86]
respiratory syncytial virus (RSV)	different mRNA species, small non-coding RNAs, nucleocapsid protein N, attachment protein G, and fusion protein F	[87]
alpha (Herpes Simplex Virus 1), beta (Human Cytomegalovirus, and Human Herpesvirus 6), and gamma (Epstein–Barr Virus, and Kaposi Sarcoma-associated Herpesvirus) herpesviruses	viral DNA, mRNAs, miRNAs, and some EBV proteins: EBV nuclear antigen-1 (EBNA-1) and latent membrane proteins 1 and 2 (LMP-1 and LMP-2)	[88,89,90]

**Table 2 cells-10-03043-t002:** Viral receptors and its potential application for treatment of human pathologies with overexpression of viral receptor.

Virus	Receptor of Fusion Protein	Localization in Normal Cells	Potential Application for Treatment of Human Pathologies with Upregulation of Viral Receptor
Influensa Mumps virus Human parainfluenza viruses	Sialic acid receptors	Ciliated epithelial cells [101,102,103]	Alteration in sialic acid processing that leads to an upregulation of sialylated glycans and its receptors in many tumors [104]
Hepatitis B virus	Sodium Taurocholate Cotransporting Polypeptide (NTCP)	Hepatocytes [105]	Use in target therapy for liver fibrosis and cancer [106]
Hepatitis C virus	CD81 tetraspanin, scavenger receptor class B type I (SR-B1)	Hepatocytes [107]	CD81 increases the progression of prostate cancer [108] High SR-B1 expression is observe in lung adenocarcinoma [109]
Rous sarcoma virus Vesicular stomatitis virus	Low density lipoprotein receptor (LDLR)	Bronchial epithelial cells [110] Epithelial cells [111]	Increased LDLR expression in Prostate cancer [112] and breast cancer [113]
Human immunodeficiency viruses (HIV)	CD209, CD4	T cells [114]	Targeting to HIV infected cells. B and T cell lymphoma [115]
Respiratory syncytial virus	IGF1R, CX3CR1	Bronchial epithelial cells [116,117]	Broad types and a range of cancers [118]
Human T-lymphotropic virus	glucose transporter-1 (GLUT-1)	T cells [119]	Broad types and a range of cancers [120]
Measles morbillivirus	CD150	Immune cells [121]	Tumors of the Central Nervous System [122]
Nipah virus Hendra virus	EphrinB2	Endothelial and smooth muscle cells in arterial vessels [123]	Uterine endometrial cancers [124]
Coronavirus	ACE2	Small intestine, testis, kidneys, lungs. [125]	Expression of ACE2 was highest in renal cell carcinoma [126]
Zaire ebolavirus Marburg virus	T-cell immunoglobulin and mucin domain 1 (TIM-1)	Kidney & urinary bladder, intestine [127]	TIM-1 overexpression in human non-small-cell lung cancer [128]
Lymphocytic choriomeningitis virus Lassa virus	α-dystroglycan	Female and Muscle tissues [129,130]	Muscular diseases treatment [131]
Lujo mammarenavirus	neuropilin-2 (NRP2)	Female and Male tissues [132]	Overexpression in breast cancer [133]
Rubella virus	Myelin oligodendrocyte glycoprotein (MOG)	Oligodendrocyte [134]	Glioma [135] Multiple sclerosis [136]
Venezuelan Equine encephalitis virus	Low-density lipoprotein receptor class A domain-containing 3 (LDLRAD3)	Neuronal cells [137]	Breast Cancer [138]
Dengue virus West Nile virus	Mannose-binding receptor (MR), CD209	Dendritic cells, Macrophage [139,140]	Gastric cancer [141]
Japanese Encephalitis Virus	PLVAP and GKN3	Dendritic cells, Macrophage [142]	PLVAP was upregulated in tumors of the brain, lungs, breasts, stomach, liver, pancreas, colon, small intestine, kidneys, ovaries, prostate, uterus, skin and lymph nodes [143]
Tick-borne encephalitis virus	Glucagon-like peptide-2 receptor (GLP2R)	Nerve cells [144]	Gastrointestinal Tumors [145]
Rabies virus	Metabotropic glutamate receptor subtype 2 (mGluR2)	Nerve cells [146]	Prostate Cancer [147] Glioma [148]
Variola virus	CD98	Epithelial cells [149]	B cell lymphomas [150]
Herpes simplex virus	Heparan sulfate (HS)	Epithelial cells [151]	Colorectal Cancer [152]
Mouse mammary tumor virus	Transferrin receptor 1	Mammary epithelial cells [153]	TFR1 is abundantly expressed in liver, breast, lung and colon cancer cells [154]
Syncytin-1	Na-dependent amino acid transporter 2 (ASCT2)	ASCT2 expression increases in highly proliferative cells such as inflammatory and stem cells [155]	Colorectal Cancer [156]

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
