# Peer review of "Viral Membrane Fusion Proteins and RNA Sorting Mechanisms for the Molecular Delivery by Exosomes"

_cells, 2021, doi:10.3390/cells10113043_

Round 1

Reviewer 1 Report

 The article discusses viral-assisted interaction between the target cell and extracellular vesicles (EVs). The article also addresses the usability of viral fusion protein in developing EV-based nanocarriers with higher efficacy of payload delivery.

The article was well-written and discuss relevant issues in the field.

Minor issues:

The author should italicized in vivo in line 19, since it is italicized other places throughout the article, such as, line 52, 305, 365.  

The author abbreviates extracellular vesicles (EVs) and should continue abbreviated it throughout. Examples including but not  limited too, lines 28, 103, 235, 285, 303, 376, 379, 391, 394, 396 (2 points), 400, 404, 415, 420.

Table 1:

The authors should include these HIV papers, if  appropriate and in line with Table #1.

Role of TIM-4 in exosome-dependent entry of HIV-1 into human immune cells. Sims B, Farrow AL, Williams SD, Bansal A, Krendelchtchikov A, Gu L, Matthews QL. Int J Nanomedicine. 2017 Jul 6;12:4823-4833. doi: 10.2147/IJN.S132762. eCollection 2017.

Tetraspanin blockage reduces exosome-mediated HIV-1 entry.

Sims B, Farrow AL, Williams SD, Bansal A, Krendelchtchikov A, Matthews QL.Arch Virol. 2018 Jun;163(6):1683-1689. doi: 10.1007/s00705-018-3737-6. Epub 2018 Feb 10.PMID: 29429034 

The author should include an adenovirus study by, Neural stem cell-derived exosomes mediate viral entry Sims et. al, 2014, if appropriate and in line with Table #1.

Table 2:

Please change title name:  add “of” between treatment and human.

In the first row, last column.. potential. add “of” between treatment and human.

In second row, last column, change “tumots” to “tumors”.

In the seventh row, last column, consider changing “Broad types range of cancer” to “Broad types and a range of cancers”.

In the eight row, last column, consider changing “Broad types range of cancer” to “Broad types and a range of cancers”.

Line 392: Change recognize to “recognition”.

Author Response

Response: 
We appreciate the reviewer for the time spent on a careful reading of our manuscript. We agree with all the proposed corrections and have made the necessary changes in the text.

Reviewer 2 Report

Authors explored an interesting topic focusing on the molecular details of the viral-assisted interaction between the target cell and extracellular vesicles. In addition, the systematic description of the usability of viral fusion proteins in developing extracellular vesicle-based nanocarriers is important.  These nanocarriers with higher efficacy of payload delivery may help others to design novel approaches for the specific and efficient delivery of the targeted molecules. Overall, this manuscript thoroughly reviews scientific findings on this important topic. Several issues that might be considered for being addressed are described below:

  1. Authors described the process of membrane fusion and specific fusion proteins/genes, which is helpful to understand the traditional membrane fusion. Meanwhile, the idea of targeting extracellular vesicles and exosomes with synthetic viral fusion proteins was also mentioned by the authors. If there is a form with comparison between the traditional membrane fusion and extracellular vesicle-based molecular transmission, such as different receptors, could help readers get a better picture.

Authors discussed the specificity of exosome targeting can be achieved via the insertion of specific integrins into membranes of exosomes, but is there a way to control the delivery of specific RNA in extracellular vesicle?  This could be discussed.

Author Response

We appreciate the reviewer for the time spent on a careful reading of our manuscript and overall positive feedback.   

Point 1: "Authors described the process of membrane fusion and specific fusion proteins/genes, which is helpful to understand the traditional membrane fusion. Meanwhile, the idea of targeting extracellular vesicles and exosomes with synthetic viral fusion proteins was also mentioned by the authors.  If there is a form with a comparison between the traditional membrane fusion and extracellular vesicle-based molecular transmission, such as different receptors, could help readers get a better picture."  

Response 1:  "If there is a form with a comparison between the traditional membrane fusion and extracellular vesicle-based molecular transmission, such as different receptors...." Although we do agree with the reviewer's suggestion, we do not have such a form of comparison. Please, note that EVs can be absorbed by the cell via endocytosis without the fusion step. Therefore, it is difficult to make a clear comparison between the traditional membrane fusion mechanism and the targeted EV fusion.  

Point 2: "Authors discussed the specificity of exosome targeting can be achieved via the insertion of specific integrins into membranes of exosomes, but is there a way to control the delivery of specific RNA in extracellular vesicle?  This could be discussed."  

Response 2:  The sorting and packaging of their RNA genomes are characteristic of various viruses. Based on the literature search, we found that there is no clear understanding of how the packaging signals work in most viruses. Only for several viruses, such as HIV1, coronavirus, and influenza virus, such mechanisms were investigated. Apparently, the specificity of RNA packaging is mediated via special packaging proteins that bind specific sequences in the target RNA. For example, the influenza virus genome is segmented into eight parts and each part has its own signal. For efficient packaging of RNA all eight signals are required, which makes the packaging signals of this virus cumbersome and difficult to use for delivery of synthetic RNAs into EVs. In the case of HIV1, the packaging signal is well understood and there is only one gag packaging protein required. The similarity between HIV and EVs biogenesis underlies the ability of the HIV genome to pack into extracellular vesicles without assembly of mature virions (doi: 10.3389/fmicb.2018.02411). Using this system, it has been shown that exogenous RNAs can be sorted and specifically packaged into EVs for subsequent delivery into the target cell.

Reviewer 3 Report

The review, "Viral membrane fusion proteins and RNA sorting mechanisms for the molecular delivery by exosomes " by Zubarev et al., describes the molecular details of the viral-assisted interaction between the target cell and extracellular vesicles.

The authors conducted a rigorous review of the literature from which they took inspiration for conclude their work hypothesize that the expanding knowledge of different mechanisms that viruses utilize to interact with host cells, together with the ability to manipulate and engineer hybrid extracellular vesicles, could be the basis for the development of future therapies.

Furthermore, the article is well written and well organized and leads the reader to an easy understanding.

For these reasons I suggest to accept the manuscript in the present form

Author Response

Response:
We appreciate the reviewer for the time spent on careful reading of our manuscript and his/her positive feedback.